# Regulatory Control and the Effects of Condensation Water on Water Migration and Reverse Migration of *Halostachys caspica* (M.Bieb.) C.A.Mey. in Different Saline Habitats

**Lu Qin** [1,2,†]**, Xuemin He** [3,4,5,†]**, Guanghui Lv** [3,4,5,]***and Jianjun Yang** [3,4,5]

1   College of Tourism, Xinjiang University, Urumqi 830046, China
2   Key Laboratory for Sustainable Development of Xinjiang's Historical and Cultural Tourism,
    Urumqi 830046, China
3   College of Ecology and Environment, Xinjiang University, Urumqi 830017, China
4   Key Laboratory of Oasis Ecology of Education Ministry, Urumqi 830017, China
5   Xinjiang Jinghe Observation and Research Station of Temperate Desert Ecosystem, Ministry of Education,
    Urumqi 830017, China
*   Correspondence: ler@xju.edu.cn; Tel.: +86-0991-2111427
†   These authors contributed equally to this work.

**Abstract:** Condensation water has been a recent focus in ecological hydrology research. As one of the main water sources that maintains the food chain in arid regions, condensation water has a significant impact on water balance in arid environments and plays an important role in desert vegetation. This study takes drought desert areas and high-salinity habitats as its focus—selecting *Halostachys caspica* (M.Bieb.) C.A.Mey. and its community in mild, moderate, and severe salinity soil—analyzed the source of condensation water utilized by these plants, and calculated its percentage of contribution. I. Study results revealed: (1) Scale-like leaves can absorb condensation water and the order of condensation water contribution to plant growth in different salinity habitats are severe > mild > moderate, such that the average contribution rates were 11.13%, 7.10%, and 3.79%, respectively; (2) The migration path of water movement in these three communities are formed in two main ways: (a) rain and condensation water recharge the soil to compensate for groundwater, while some groundwater compensates for river water and partially returns to the atmosphere by soil evaporation and plant transpiration; and (b) rain and condensation water directly compensate for river water and plant roots absorb river water, groundwater, and soil water in order to grow; (3) in mild habitats, the water movement path in plants is as follows: shallow root → stem → branches → leaves and shallow root → deep root; (4) in moderate habitats, stems act as the bifurcation point and the path follows as: stem → branches → leaves and stem → shallow root → deep root; and (5) in severe habitats, the path is as follows: deep root → shallow root → stem → branches → leaves, and finally returning to the atmosphere. These results elucidate the contribution of condensation water on *Halostachys caspica* growth and the migration path through the *Halostachys caspica* body. Condensation water obtained by *Halostachys caspica* communities in different salinity habitats provides a theoretical basis and data supporting the need for future research of condensation water on plants at the physiological level in arid regions and provides reference for the protection of saline soil and its ecological environment in arid regions.

**Keywords:** condensation water; isotope; *Halostachys caspica*; salinity; moisture migration

## 1. Introduction

In arid regions, precipitation is limited and evaporation is intense, and any supplementary water has a positive impact on the ecosystems. Condensation water and precipitation are two water sources in desert regions that play an important role in desert ecosystems [1–3]. In the desert, water is scarce and apart from precipitation, condensation

water is an important, vital source of water. In drought conditions, there is less soil water content and fewer perennial plants. Despite the small volume of condensation water, it plays an important role in the local water balance [4–6], and especially in drought years its importance is more obvious [5,7].

In the abundant precipitation region, the amount of condensation is trivial. However, in the arid and semi-arid regions, condensation water plays a supplementary role (e.g., in the desert along the Atlantic coast) [8–10]. Maphangwaa et al. [11] found that atmospheric water vapor is the main water source that lichen absorb, and it is atmospheric water vapor, not precipitation, that determines the amount of lichen richness and overall distribution. Thus, conducting research on condensate water and the water balance in arid regions should be a focus of present and future studies.

In a former study conducted by Tao and Zhang [12], it was demonstrated that the tippy of desert moss crust can significantly reduce and delay evaporation in the crust, which prolongs plant hydration time. The greater the amount of precipitation, the more obvious the slowing effect is observed in the tippy of desert moss crust. This slowing effect is also helpful as it allows for the utilization of condensation and precipitation, enhancing the moss crust's ability to adapt to drought conditions. In a separate study, Temina and Kidron [13] discovered that the duration of condensate water determines the distribution of lichen on rocks in the Negev desert. Cheng et al. [14] found in the Alpine Sandy Desert that biological soil crust is conducive to moisture absorption and condensate water and, with the development of the crust, the content of water vapor increases. As an important source of water, condensation water plays a significant role in the survival of vegetation in arid areas [15,16]. Therefore, when conducting research on water balance in arid regions, condensate water cannot be ignored and the importance of water vapor on desert vegetation has been a recent focus of present-day studies.

There are two aspects of plants' use for condensation water. One is indirect utilization, that is, when the dew congeals on the surface of the plant or when fog is intercepted by the canopy and as a result, small water droplets form on a big plant's leaves and water drips down to the soil surface where it is absorbed by the root system of shallow root plants. This use of condensation water allows scholars to focus on the issues of water sources for root systems, such as groundwater, soil natural water, precipitation, fog, dew, and more. Thus, the contribution of various sources of water supply for a specific plant can be calculated, mainly for water sources that contain different $\delta D$ and $\delta^{18}O$ isotopes to distinguish various sources. For example, Goebel and Lascano [17] conducted a quantitative analysis of the water use conditions of cotton, including the utilization of condensation water by measuring the different sources of water containing $\delta D$ and $\delta^{18}O$.

Additionally, when plants absorb water, it affects the direction of moisture migration through the plant body, but research on this characteristic is rare. Previous research focused on the quantity of water in terms of soil condensation, plant canopy condensation, occurrence regularity, and impact factors. There are very few studies on the quantity of condensation water utilized by plants and moisture migration path through the plant body [18], and the signature of $^{18}O$ isotope tracer (the phenomenon of the enrichment of $^{18}O$ in the photosynthetic organs, secondary branches, and trunk xylem) showed that the photosynthetic organs of desert woody plants are able to absorb canopy dew and transfer it to the trunk xylem. In high humidity conditions, assimilating branches of *Haloxylon ammodendron* (C.A.Mey.) Bunge actively absorbed canopy dew and transferred the canopy dew down to the secondary shoots through the reverse water potential ($\Psi$) gradient between photosynthetic organs and secondary branches ($\Psi$ Photosynthetic organs > $\Psi$ Secondary branches), and the photosynthetic organs can transport the excess dew to the trunk stem via reverse water potential gradient, which is conducive to the continuous absorption and utilization for canopy dew [18]. Leaves of desert plants usually carnify and degenerate to form spiny or scaly parts, while succulent leaves can effectively store water but reduce water transpiration by reducing the amount of leaf area exposed to the air [19]. *Halostachys caspica* is a model organism that belongs to the saline-xerophytic-juicy subshrub and is an

important windproof, dune-fixing shrub, whose branchlets contain succulent juice and has scale-like leaves. Studies have shown that irregular leaf surface can capture small water droplets, and the leaves of *Halostachys caspica* have an irregular surface; thus, we can infer that *Halostachys caspica* could absorb and utilize condensate water.

In this study, we test the following three hypotheses: (1) The leaf of *Halostachys caspica* has the ability to absorb condensation water; (2) The contribution of condensation water to plant growth in *Halostachys caspica* habitats with salinity differences; and (3) The different ways in which condensation water is utilized by *Halostachys caspica*. This study selected the typical desert shrub species, *Halostachys caspica*, as the research organism, which is found in the Ebinur Lake wetland national nature reserve. The treatments were divided into three different salinity habitats: mild, moderate, and severe salinity soils. We conducted a field investigation and analyzed isotope composition to determine the source of condensation water that plants used, as well as its proportion, and investigated the migratory path of condensation water in the plant body. The findings of this study will help to further our understanding of the contribution of condensation water and the migration path in the *Halostachys caspica* body under different salinity conditions and to probe the migration path of obtained condensation water in the *Halostachys caspica* body. The results help to further understand the contribution and the migration path in the *Halostachys caspica* body to the obtained condensation water by the *Halostachys caspica* communities under a different salinity habitat. It is of great significance to elucidate the positive effects of condensation water on plant growth in desert ecosystems in order to provide a theoretical basis and data support for propelling the future research of condensation water on plant physiological level in arid regions and to provide reference for the protection of saline soil and its ecological environment in arid regions.

## 2. Materials and Methods

### 2.1. The Study Region

The study region is located on the northwest edge of the Junggar Basin (82°36′–83°50 E, 44°30′–45°09′ N) of the Xinjiang Uygur Autonomous Region. The climate is very dry, there is scarce rainfall, copious amounts of sunlight, strong winds, dust storms, hot summers, and cold winters, which is typical of climates belonging to temperate continental arid regions [20]. Influenced by landform characteristics and climactic conditions, the distribution of vegetation within the Ebinur Lake basin is primarily dominated by two plant flora native to Asia and Mongolia. These two plants had a clear transition into these areas and represent most desert plant species in Xinjiang. *Halostachys caspica* is one of the dominant species in the study area and it is a saline, xerophytism, succulent subshrub that is mainly distributed in the salt-alkali beach area, river valley, and by the salt lake.

Salinization reverse succession was the main method utilized in this study. In the study area, we selected communities for mild, moderate, and severe salinity (Figure 1), and their soil salt content were 0.500 ± 0.275%, 1.428 ± 0.286%, and 2.022 ± 0.329%, respectively. The mild salinity community is adjacent to the river, located at the north of the river with a vertical distance from the river of 50 m. The moderate salinity community is far away from the river bank, located at the north of the river with a vertical distance from the river of 3700 m. The severe salinity community is far away from the river bank, located south of the river with a vertical distance from the river of 700 m. Three well-growing *Halostachys caspica* were selected for isotope sampling in each habitat.

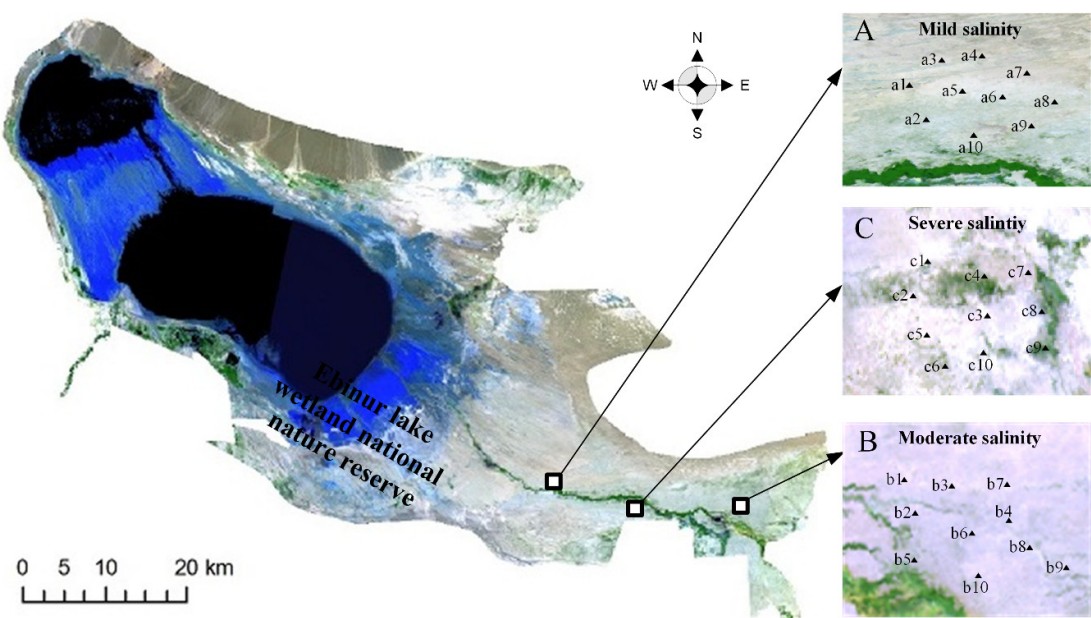

**Figure 1.** Schemes of study sample regions.

### 2.2. Experimental Method and Samples Collection

This article had set seven water sources, which were: topsoil (0–5 cm), shallow soil (5–20 cm), deep soil (20–30 cm), rain water, river water, ground water, and condensation water.

Plant samples were collected before sunrise and three species of plants were randomly selected as representatives of either the mild, moderate, or severe salinity habitats; there was a total of nine samples. We used loppers to acquire leaves. Suberification was utilized after gathering mature branches, stems, shallow roots (5–20 cm), and deep roots (20–50 cm) in order to remove the phloem but leave in the xylem. We then quickly put samples into 50 mL isotope sampling bottles and sealed them with parafilm membrane. Samples were transported back to the lab, then placed in the fridge to freeze samples at −20 °C until they were analyzed for isotope composition.

Soil samples were collected before sunrise from each of the three salinity habitats. Samples were gathered under the canopy of plant samples, close to the root. Samples were collected from the surface soil (0–5 cm), shallow soil (5–20 cm), and deep soil (20–30 cm). We then placed the samples into 50-mL isotope sampling bottles and sealed them with parafilm membrane. Samples were transported back to the lab, then placed in the fridge to freeze samples at −20 °C until they were analyzed for isotope composition.

Natural water samples were collected from the underground water source near the study point. To ensure the samples were true representatives, we collected deep water samples from the Aqikesu River. Rain and condensation water samples were collected two days before plant and soil sampling, then placed in a 50 mL isotope sampling bottle and sealed with parafilm membrane. Samples were transported back to the lab, then placed in the fridge to be kept at 4 °C until they were analyzed for isotope composition.

Using a handheld weather instrument (Kestrel 4500 NV), we were able to determine the air temperature, dew-point temperature, relative humidity of the atmosphere, wind speed, and air pressure at 1.5 m above the surface when all of the samples were collected.

### 2.3. Plant Physiological Determination

#### 2.3.1. Presence and Absence Condensation Water Treatment Setting

In the mild salinity, moderate salinity, and severe salinity communities, some leaves with the same water potential were selected in the observation plot approximately 0.5 h before sunset, and they were divided into three groups, namely: (1) Before treatment (B), the initial water potential of branches were measured immediately after taking down the

branches; (2) presence of condensation water treatment (W1), that is, branches were not bagged in the natural state; (3) absence of condensation water treatment (W0), sealed with plastic bags (so that the surface of branches was not affected by night condensation water).

### 2.3.2. Water Potentials

Approximately 0.5 h after sunrise the next day, the branches were removed to check the presence and absence of condensation water, and the water potential of the branches was immediately measured, with 9 repetitions in each group.

### 2.3.3. Fresh Leaf Water Absorption per Unit Area

In mild salinity, moderate salinity, and severe salinity communities, six leaves with and without condensate treatment were picked, the radius (*r*) and length (*l*) of the leaves were measured, and the weight *m*1 was weighed. Then, the leaves were placed in clean water and left for approximately 24 h. After the leaves fully absorbed water, they were removed from the water. The water on the surface of the leaves was completely absorbed by filter paper, and *m*2 was weighed again. The water absorption per unit area of fresh leaves was calculated by the following formula:

$$\text{Fresh leaf water absorption per unit area} = (m2 - m1)/(\pi r^2 l) \tag{1}$$

### 2.3.4. Transpiration Rate (Tr) and Water Use Efficiency (WUE)

The transpiration rate and the net photosynthetic rate (Pn) of *Halostachys caspica* under the presence and absence condensation water treatment were measured by cluster leaf chamber of Li-6400XT portable photosynthetic apparatus (Li-Cor, Lincoln, Nebraska, USA), and the WUE was calculated by the following formula:

$$\text{WUE} = \text{Pn}/\text{Tr} \tag{2}$$

In the presence and absence condensation water treatment, 30 repetitions are determined for each treatment.

### 2.4. Water Extraction and Isotope Determination

We used the Liquid Water Isotope Analyzer (LWIA, DLT–100, Los Gatos Research Inc., Mountain View, CA, USA) to determine the hydrogen and oxygen isotopic composition of the samples. Plant water and soil water were extracted using a cryogenic vacuum distillation line [21] and the extracted water samples were stored in sealed glass vials at 2C. Then, the hydrogen and oxygen isotopic compositions of the samples were determined by an isotope ratio infrared spectroscopy (IRIS) analyzer—the Liquid Water Isotope Analyzer (LWIA, DLT–100, Los Gatos Research Inc., Mountain View, CA, USA), and the determination of each sample was repeated 12 times. Analytical precision of individual measurement were $\pm 0.1\permil$ for $\delta$D and $\pm 0.25\permil$ for $\delta^{18}$O. The isotopic composition can be expressed as:

$$\delta\text{X} = \left( \frac{R_{sample}}{R_{standard}} - 1 \right) \times 1000\permil \tag{3}$$

where X is D or $^{18}$O, $R_{sample}$ and $R_{standard}$ are the hydrogen or oxygen isotopic composition ($^2$H/$^1$H or $^{18}$O/$^{16}$O molar ratio) of the sample and the standard water (Standard Mean Ocean Water, SMOW), respectively.

### 2.5. Data Analysis

A one-way analysis of variance (ANOVA) was utilized to analyze the data. Data manipulation and visual representations were conducted using Excel 2003 software and Sigmaplot 10.0. All of the statistical analyses were run with a significance level of 0.05 using StatView 5.0 (SAS Institute, Inc., Cary, NC, USA). There were seven sources of water in this

study, according to the upper and lower limits method Proposed by Phillips and Gregg [22]. IsoSource 1.3.1 software was used to analyze the water source of plant and soil samples.

## 3. Result and Analysis

### 3.1. Plant Physiological Characteristics in Different Salinity Habitats

#### 3.1.1. Water Potentials

As can be seen from the Figure 2, night condensation water can improve the water potential of plant leaves in those three communities, but without condensation water supplement, it will reduce the water potential of plant leaves, indicating that the leaves of *Halostachys caspica* can absorb condensation water and improve the water potential of plant leaves.

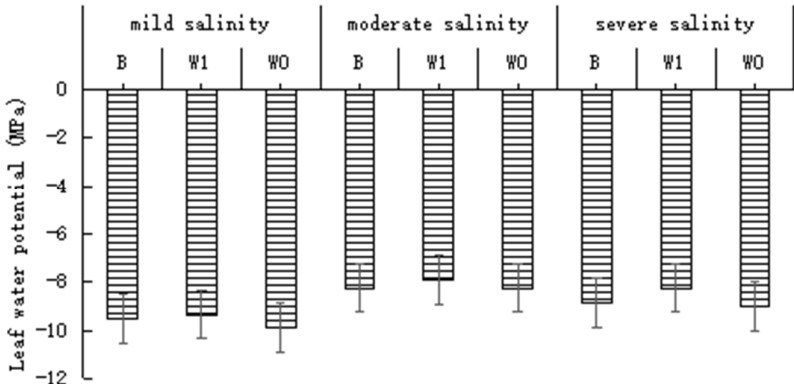

**Figure 2.** Comparison of *Halostachys caspica* leaves water potentials under presence and absence of water treatment in different salinity habitats. Before treatment (B), presence of condensation water treatment (W1), absence of condensation water treatment (W0).

#### 3.1.2. Fresh Leaf Water Absorption per Unit Area

Using the results of fresh leaf water absorption per unit area in mild salinity, moderate salinity, and severe salinity communities under the presence and absence condensation water treatment, the following figure can be drawn.

In mild salinity, moderate salinity, and severe salinity communities, the fresh leaf water absorption per unit area of leaves of absence condensation water is higher than that presence condensation water (Figure 3), which indicates that the short-term lack of water on the surface of leaves can stimulate the water absorption per unit area. In addition, under the condition of absence condensation water treatment, the water absorption per unit area of leaves of *Halostachys caspica* in the mild salinity, moderate salinity, and severe salinity communities showed the following order: moderate salinity community > severe salinity community > mild salinity community, which indicated that if the leaves were temporarily short of water, and once there was enough water in the air, and the salinization degree increased, the plant leaves growing on them would have stronger water absorption capacity.

Under the presence and absence condensation water treatment, the transpiration rates of three communities are as follows: severe salinity community > moderate salinity > mild salinity community (Figure 4), which indicated that the ecological effect of condensation water on severe salinity community was higher than that of mild salinity and moderate salinity community. In those three salinity habitats, the transpiration rate of plants treated of presence condensation water was higher than that absence condensation water, which indicated that condensation water can improve the transpiration rate of plants and had certain ecological effects on plants. In addition, it was found that the condensation water formed at night can supplement the evaporated water to a certain extent in the three communities [23].

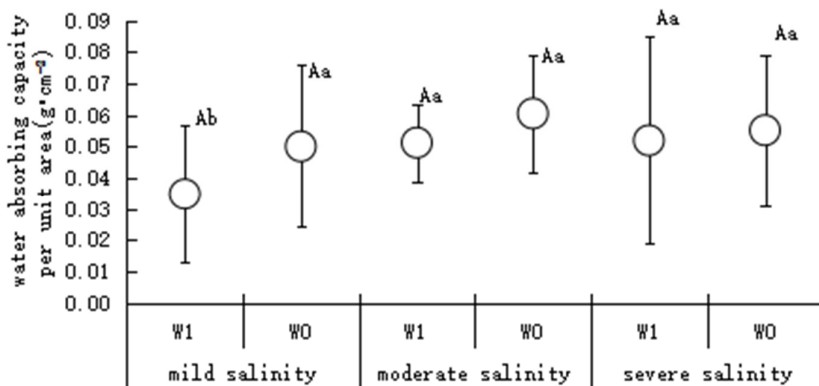

**Figure 3.** Fresh leaf water absorption per unit area under presence and absence of condensation water treatment. The same lowercase letter indicates that there is no significant difference among different treatments in the same habitat ($p > 0.05$). The same capital letter indicates that there is no significant difference between different habitat of the same treatment ($p > 0.05$), presence of condensation water treatment (W1), absence of condensation water treatment (W0), the same below.

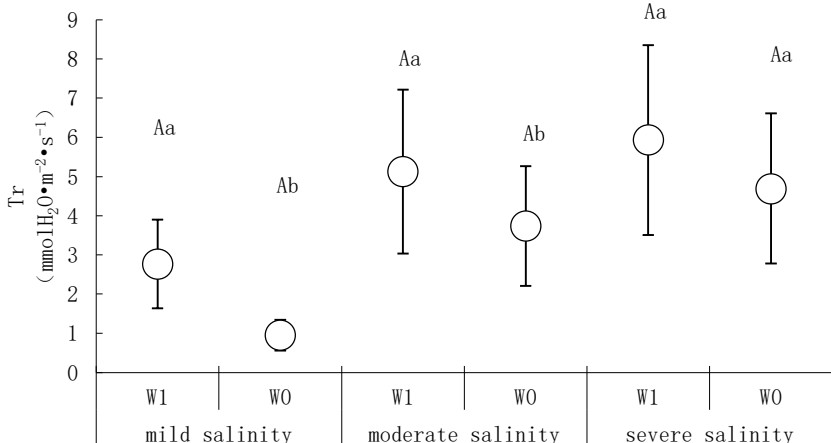

**Figure 4.** Tr of *Halostachys caspica* under the presence and absence condensation water treatment.

The instantaneous water use efficiency (WUE) of plants is a comprehensive physiological index to evaluate the adaptability of plants to the environment.

Daily mean value of the WUE of moderate and mild salinity communities under the presence condensation water treatment higher than the absence condensation water (Figure 5), which may be temporary water balance is out of balance in plants at night without condensation water supplement, which reduces the water use efficiency. In the severe salinity, the daily average WUE of *Halostachys caspica* under the presence condensation water treatment is lower than that of absence treatment. This is because the severe salinity community is far from the water source, lives in a low water environment for a long time, has a survival strategy, has a strong self-regulating mechanism for water stress to improve water utilization ability to cope with water deficit [24], which is similar to Yu's research result for maize [25].

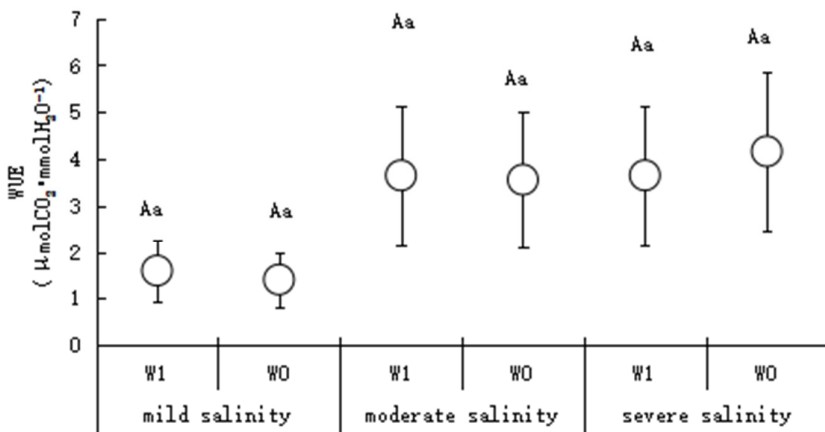

**Figure 5.** Comparison of *Halostachys caspica* WUE under presence and absence of condensation water treatment in different salinity habitats.

### 3.1.3. δD and δ¹⁸O Composition in Plant Structures and the δD-δ¹⁸O Relationship in Different Salinity Habitats

Plant leaves enriched with δD and δ¹⁸O isotopes have partially negative δD values ranging from −40.0844 ± 1.6106 to −40.0844 ± 1.6706%; when there are partially positive δ¹⁸O values, they range from 4.9758 ± 0.686 to 12.469 ± 1.653% (Figure 6). The max value was observed in the severe salinity community, while the minimum value was observed in the light salinity community, which indicates that *Halostachys caspica* growing in severe salinity soil were better at evapotranspiration. Additionally, compared to aerial parts of the plant, such as branches and stems, the shallow roots from moderate salinity soil had lower δD and δ¹⁸O values. This may be because soil evaporation has allowed for the transpiration and evaporation at the roots of the plant. Moreover, transpiration and evaporation in aerial parts of the plant strengthen water delivery from the roots to aerial parts of plant, which suggests that there is a regulatory effect to the plant from soil moisture through a hydraulic redistribution effect [26].

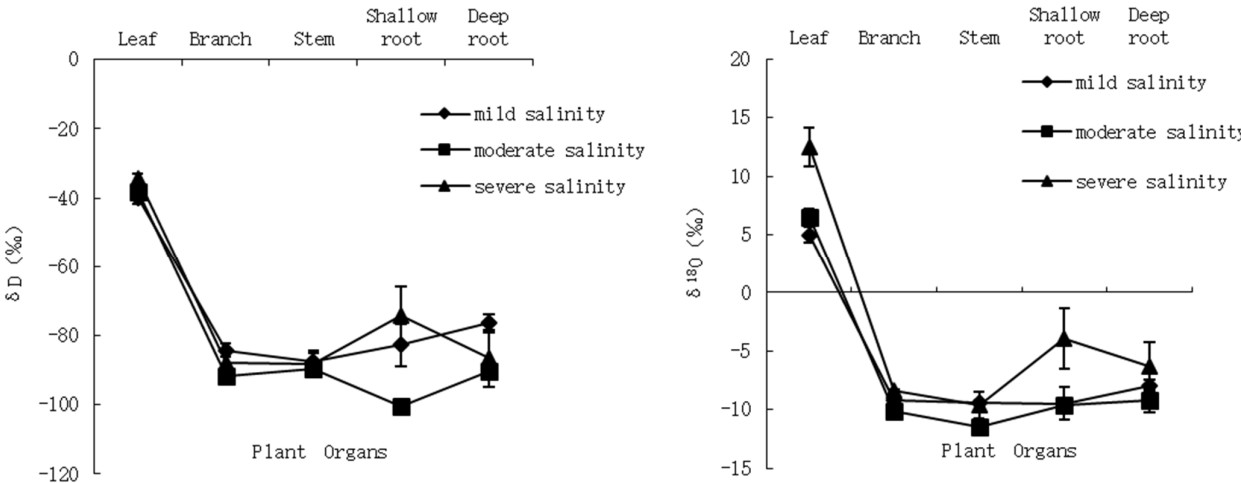

**Figure 6.** Distribution of δD and δ¹⁸O composition of structures in different salinity habitats.

The three salinity habitats were ranked as mild > moderate > severe, which indicates that the degree of soil salinity enhances the evapotranspiration ability of *Halostachys caspica* (Figure 7). This in turn causes an isotope fractionation effect. It could also be a survival strategy for plant growth in saline soil is to take excess salt from the plant of body as evapotranspiration increases.

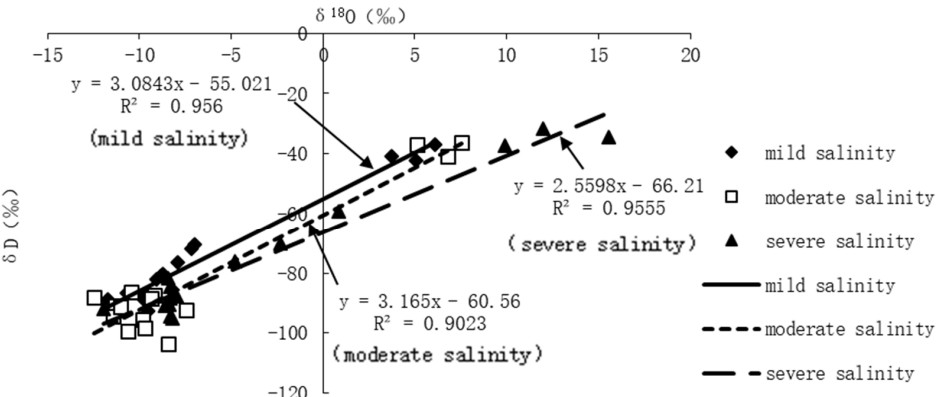

**Figure 7.** δD-δ$^{18}$O relationship in water samples from plant structures found in different salinity habitats. Fitting equations for each habitat are noted in the figure.

*3.2. Hydrogen and Oxygen Isotope Composition and Soil Profiles of Halostachys Caspica Communities*

Results revealed that there are significant differences in δD and δ$^{18}$O values based on the salinity habitat (Figure 8). The values were $(-70.524 \pm 4.326)$–$(-46.432 \pm 3.638)$‰ and $(-5.197 \pm 0.906)$–$(1.987 \pm 0.626)$‰, respectively. The higher the composition of δD and δ$^{18}$O isotopes, the more enriched the soil surface layer and with the increasing soil depth, the δD and δ$^{18}$O values decrease gradually.

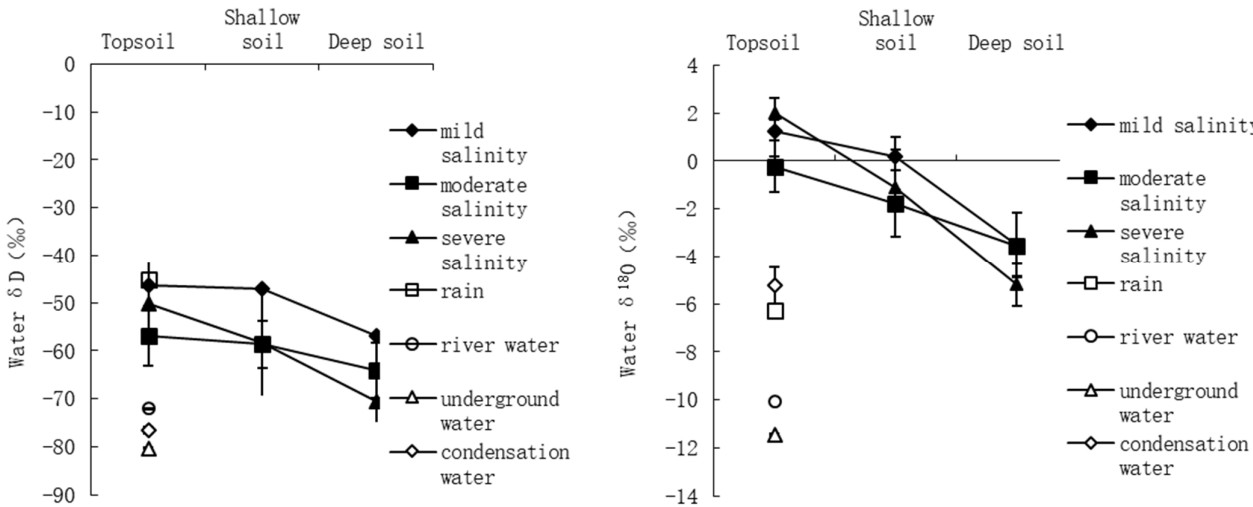

**Figure 8.** δD and δ$^{18}$O of soil water in different soil and natural water of salinity habitats.

All of the soil layers were enriched with δD in the mild salinity habitat. The topsoil in severe salinity habitats had slightly higher δ$^{18}$O values than the mild salinity, while δ$^{18}$O values in mild salinity soil water were higher than moderate and severe, which indicates that there is soil evaporation and an isotope fractionation effect in mild salinity, although the volatility of δD and δ$^{18}$O values are at a minimum. Meanwhile, soil water hydrogen and oxygen stable isotopes are lacking in deep soil of severe salinity habitats.

Results revealed that closer to the surface, δ$^{18}$O values for soil water are partially positive. Additionally, Δ δD values for soil water in the different salinity habitats were between the values for rain and the other three sources of water (river water, condensation water, and groundwater). Lastly, the δ$^{18}$O values were all partially negative for all water sources.

### 3.3. The Contribution of Condensation Water to Plant Growth

Due to strong fractionation in the leaves, there were no results for leaf water source, so we analyzed the water source for branches, stems, and shallow roots. Water sources were topsoil, shallow soil, deep soil, rain water, river water, groundwater, and condensation water (Table 1).

**Table 1.** Sources of water in different salinity habitats (mean ± SE).

| Habitat | Contribution of Water Source (%) | | | | |
|---|---|---|---|---|---|
| | Soil Water | Rain | River Water | Ground Water | Condensation Water |
| mild salinity | 11.375 ± 2.071 | 8.525 ± 1.280 | 27.025 ± 0.330 | 46.1 ± 4.594 | 7.05 ± 1.100 |
| moderate salinity | 6.775 ± 2.507 | 4.575 ± 1.661 | 18.5 ± 5.750 | 66.4 ± 11.253 | 3.775 ± 1.375 |
| severe salinity | 27.9 ± 9.484 | 11.925 ± 2.258 | 20.1 ± 3.955 | 28.975 ± 10.150 | 11.125 ± 2.478 |

Results revealed that the main contributors to plant growth in mild salinity habitats were soil water (11.375%), rain (8.53%), river (27.03%), groundwater (46.1%), and condensation water (7.1%). In the moderate salinity habitats, the main contributors were 1.8% (Topsoil), 2.18% (Shallow soil), 2.8% (Deep soil), 4.58% (Rain), 18.5% (River water), 66.4% (Ground water), and 3.78% (Condensation water). In the severe salinity habitat, the main contributors were 27.91% (Soil water), 11.93% (Rain), 20.1% (River water), 28.98% (Ground water), and 11.13% (Condensation water).

To summarize, results revealed that groundwater is the main contributor to plant growth in *Halostachys caspica* with the maximum contribution in the moderate salinity community (66.4%) followed by the mild salinity community (28.89%) and the minimum contribution in the severe salinity community (28.98%). Additionally, there were multiple water absorption strategies exhibited in *Halostachys caspica* growth, which ensures that when there are water restrictions other strategies can instead be employed. Lastly, the contribution of condensation water to plant growth in severe salinity habitats was the largest, with an average value of 11.13% and a maximum of 82%. In the mild salinity habitat, the average value was 7.1% and the maximum value was 56%. In the moderate salinity habitat, the average value was 3.79% and the maximum value was 36%.

### 3.4. Water Migration Path in the Plant Body

Generally speaking, when water is transported in a mature plant, hydrogen and oxygen stable isotope fractionation will not occur. Only the leaves or high salinity plant will have fractionation, thus, one can use hydrogen and oxygen stable isotope technology to quantify these compositions in the vast majority of land plants. The $\delta^{18}$O isotope is present at the earliest water source, thus, this value will gradually increase over time and the moisture migration path in the body of *Halostachys caspica* in different salinity habitats can be determined (Figure 9a–c).

In all three habitats, rain and condensation water directly contributed to the river water and supplied the soil with water through soil infiltration. Groundwater also supplies the river with water and through soil evaporation and transpiration into the atmosphere, plant roots are able to absorb river water, groundwater, and soil water, which all contribute to plant growth.

All of the plant structures, except leaves, had $\delta^{18}$O values similar to soil water values. Yang [27] also observed this in *Haloxylon ammodendron* at the study area and speculated that in addition to rain, river water, groundwater, and soil water, *Haloxylon ammodendron* may take advantage of condensation water through strong evaporation through their leaves as a result of day and night temperature differences. Moreover, Yang [27] calculated that the contributions of condensation water to plant growth were 7.1% (mild salinity), 3.78% (moderate salinity), and 11.13% (severe salinity), which indicates that the contribution of

condensed water to plants in different saline habitats is influenced by the soil properties of saline habitats.

In the plant body, water does not simply migrate from the roots to the leaves then return to the atmosphere through evaporation. Rather, in *Halostachys caspica* communities under mild salinity conditions, water migration flows as follows: shallow roots → stems → branches → leaves; there are also short water circuits from shallow roots to deep roots. In moderate salinity conditions, the stems acts as a bifurcation point and one path of water migrates from: stems → shallow and deep roots, while the other path flows from: stems → branches → leaves. In severe salinity conditions, water migrates from: stems → shallow and deep roots →branches → leaves; there is also a water loop from the stems to shallow roots.

In terms of soil water, there are different $\delta^{18}O$ compositions in topsoil and shallow soil water along the vertical section. Shallow soil water $\delta^{18}O$ composition showed a partially negative trend from bottom to top under the topsoil, which indicates that there is an ascending phenomenon. In mild and moderate salinity conditions, topsoil water and shallow soil water $\delta^{18}O$ composition under the topsoil showed a partially positive trend up to the surface. This may be because mild and moderate salinity habitats are near the river and, thus, the soil surface absorbs more precipitation [23].

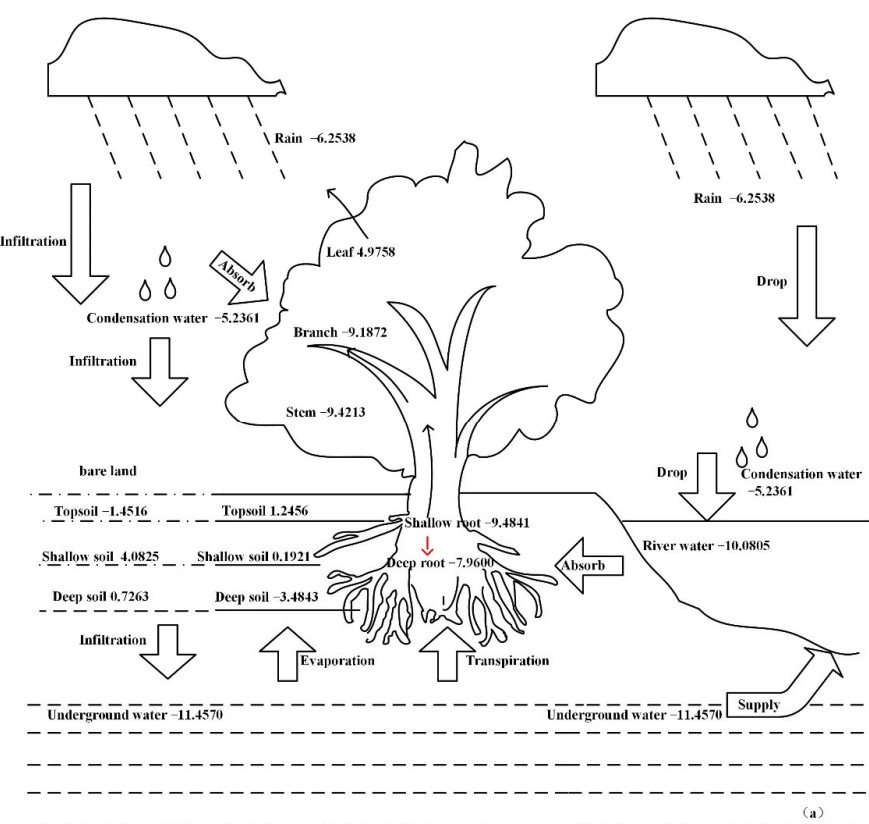

**Figure 9.** *Cont.*

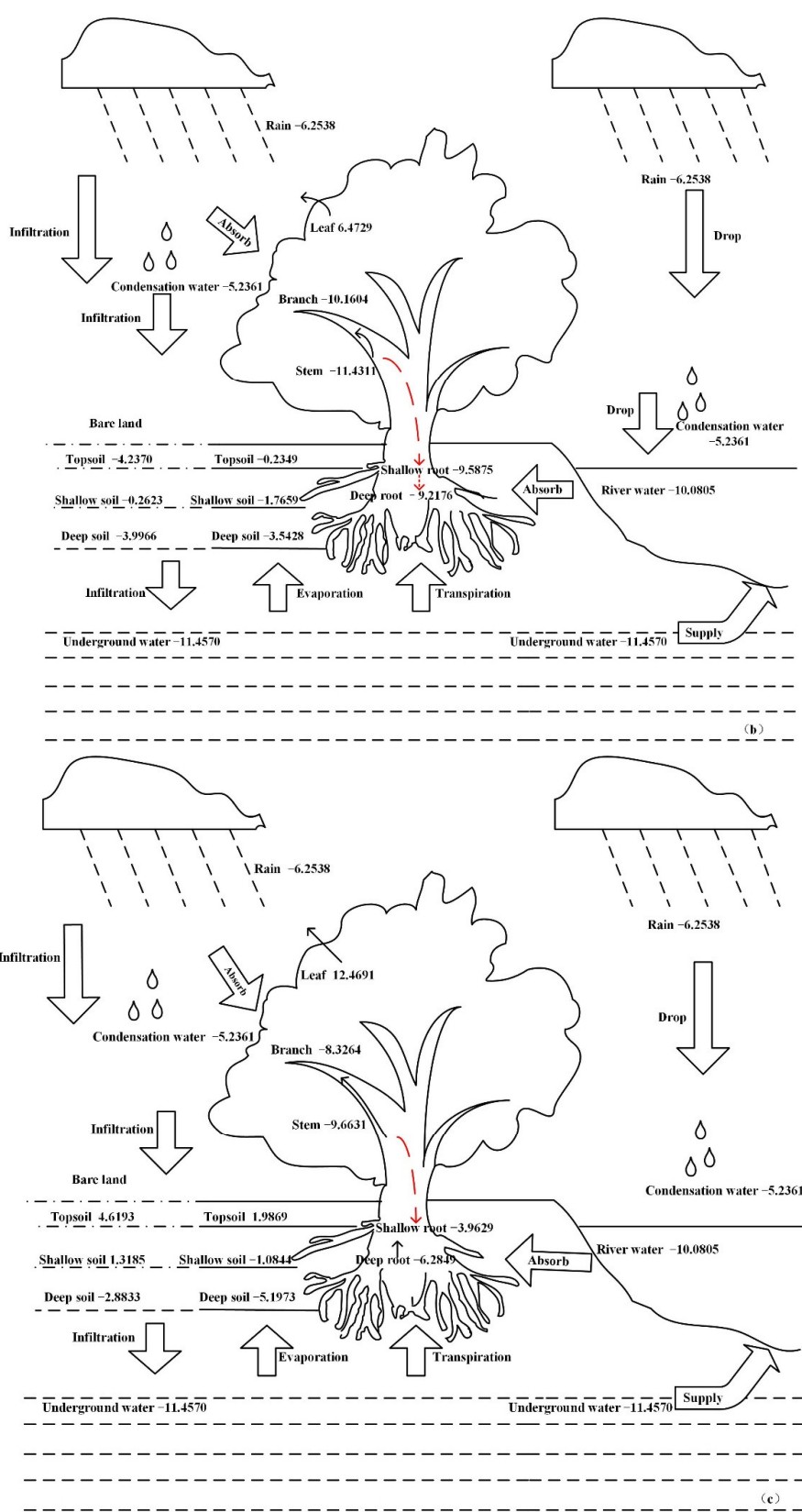

**Figure 9.** (**a**) Soil-vegetation-atmosphere water migration diagram in a mild salinity community. (**b**) Soil-vegetation-atmosphere water migration diagram in a moderate salinity community. (**c**) Soil-vegetation-atmosphere water migration diagram in a severe salinity community. Note: the unit of $\delta^{18}$O is presented as percentages. The arrow means the direction of water transport.

## 4. Discussions

### 4.1. The Contribution of Condensation Water to the Water Needed for the Growth Halostachys Caspica

In the wild, the water that plants utilize come from precipitation, soil water, runoff (including melting snow and ice), and ground water [28]. In arid regions, there is little precipitation, so groundwater is the only water source available for plant survival, especially perennial plants [29,30]. Using hydrogen and oxygen stable isotopes to study the water sources of plants, we quantitatively analyzed the origin and direction of water and the selective absorption and water use in three ecological environments that differed in salinity levels. Groundwater was determined to be the main water source utilized for plant growth by *Halostachys caspica* in different salinity habitats found within the study area. The utilization of ground water in the mild, moderate, and severe salinity habitats accounted for 46.1%, 66.4%, and 28.98% of the total water use, respectively. Thereunto, the severe salinity plants relied on groundwater the least, followed by the mild habitat, while the moderate habitat was the highest, such that groundwater accounted for more than half of all water sources. Therefore, it is clear that the change of groundwater level has had the greatest influence on plant growth in the moderate salinity area.

Although groundwater is the main water source, soil water is probably the most important source of water for these plants. Due to the nature of the soil (i.e., particle size and porosity), there are differences in the composition of hydrogen and oxygen stable isotopes. In general, surface soil water is able to contact surface moisture, thus, there is relatively large evaporation intensity to produce hydrogen and oxygen stable isotopic fractionation; this is based on established theories that state there is "light" molecular priority to evaporate ($^{1}H^{1}H^{16}O$) compared to deeper soil water levels. Surface soil water is more enriched with heavier hydrogen and oxygen stable isotopes, so there are significant differences in the levels of hydrogen and oxygen stable isotopes. Soil profiles taken at different depths show that soil water hydrogen and oxygen stable isotopic composition are indeed different, such that there is relatively stable isotopic content up to the deepest layer [26,31]. Moreover, non-moving water was a result of evaporation close to the surface area [32].

The results of this study also revealed that the closer to the soil surface, soil water $\delta^{18}O$ levels are partially positive, which is similar to the findings of other studies [33]. This is because the light isotope of oxygen evaporates, which leads to $^{18}O$ relative enrichment. As for $\delta D$ values, soil water in the three salinity habitats had $\Delta\delta D$ values between those of rain and the other three natural water bodies (river water, condensation water and groundwater). This indicates that soil water comes primarily from rain, river water, condensation water and groundwater. Furthermore, soil water $\delta^{18}O$ values were all partially negative relative to the natural water bodies and were similar to condensation water ($-5.23614 \pm 0.72364\%$). This further illustrates that soil water in the study area comes from condensation water. These results are similar to the findings of Zhu and Jiang [34], who conducted a study in the northern Loess Plateau. Namely, they found that atmospheric condensation water contributed between 0–20 cm to soil water, and that it is difficult to contribute to the soil water at the 30 cm soil layer and below.

Some scholars employ the use of thermal pulse technology, isotope tracer techniques, fluorescent tracer methods, thermal ratio, botanical anatomy, pressure chamber, and physiological measurements [35] and found that plant leaves absorb water to ease water deficit in the body of the plant through the leaf stomata, bristles, fissure, drainage organs, and other structures on the surface of leaves [6,36–38]. In a study conducted by Burgess et al. [39], isotopic tracer techniques were used on the leaves of *Sequoia sempervirens* located in the United States and found that their main source of water comes from the fog formed overnight. Ellsworth and Williams [40] conducted a study on 16 species of drought and semi-arid shrubs, and one species of mesophytic herbage kept under control conditions. They found that due to transpiration processes, there was a $\delta D$ and $\delta^{18}O$ enrichment found in leaf water, which is likely the source of water for young stems. Zhuang and Zhao [38] also conducted a study on whether the leaves of desert coat plants, *Bassia dasyphylla*, and glabrous plants,

*Agriophyllum squarrosum* (Linn.) Moq., absorbed condensation water. Results revealed that leaves of desert plants can absorb condensation water and through the process of photosynthesis, water retention and growth all respond to the presence of condensation water. Meanwhile, glabrous plants could not absorb condensation water. Vitarelli et al. [41] also conducted research on croton plants and demonstrated that the trichoid structure is the key structure through which plants absorb atmospheric water. Yan et al. [42] used a molecular ecology approach to research the intrinsic mechanism of the leaves of *Tamarix ramosissima*, a desert woody plant that absorbs condensation water from the canopy. Yang et al. [43] studied short-life desert plants and found that the leaf and stem can absorb condensation water, and increasing amounts of condensation water can significantly affect population dynamics. Cen and Liu [44] conducted research on the effects of simulated condensation water on the physiological characteristics of the leaf surface structure of *Leymus chinensis* and *Agropyron cristatum* under drought conditions and found that the aboveground biomass and root biomass increased with the presence of condensation water, and that condensation water can both protect and repair damage on plant leaf surface structures following stress induced by drought conditions. In this study, the leaf of *Halostachys caspica* could absorb the condensation water, this is because that hydrophilic polysaccharide compounds are found in cell walls of leaf epidermal cells, eutrophic cells and vascular bundle cell for photosynthetic organs of plants in arid region, these polysaccharides connect an extracellular network within the photosynthetic organs, and accept the moisture absorbed by the cuticle and transport it to the xylem. The microstructure on the surface of photosynthetic organs and the hydrophilic compounds inside which are the material basis of the canopy could absorb the condensation water [18]. In addition, In photosynthetic organs, the aquaporins (AQPs) regulating the plasma membrane and vacuolar membrane plays an important role in the process of transporting the condensation water in the canopy among cells through the symplastic pathway in desert woody plants, which is the molecular mechanism that absorbs the condensation water in the canopy [18].

The contribution of condensation water on plant growth in *Halostachys caspica* should not be overlooked, and the degrees of condensation water use vary under different salinity conditions. The degrees of condensation water use in mild, moderate, and severe salinity habitats were 7.1%, 3.78%, and 11.13%, respectively. This is roughly equivalent to the contribution of rain water in these habitats, which was 8.53%, 2.18%, and 11.93%, respectively. The degrees of condensation water use was relatively high. This may be because the surface soil salinity level was also high, which allows for greater absorption of condensation water, which infiltrated the soil where plant roots were then able to absorb and use the water for growth. Furthermore, salt crusts can reduce soil moisture evaporation, allowing more soil condensation water to infiltrate it. It was also found in this study that the degree of isotopic fractionation was positively correlated with salt tolerance; namely, the salt tolerance of plants lead to the fractionation of hydrogen isotopes [40].

### 4.2. The Water Migration Path in Halostachys Caspica

Under normal conditions, there are two paths of water migration in plants. First, roots transport water through the root system so that the cells in the soil absorb moisture from the soil through osmosis and root hair cells absorb soil moisture from the soil to the root hair cell through osmosis, which then transfer moisture to the plant root catheter through the intercellular osmotic differential and this in turn transfers moisture to each plant structure on the ground. Second, plant cells above the ground absorb moisture when guard cells on the plant leaves open, which allow a small amount of water vapor from the atmosphere to be absorbed by the plant tissue.

The water migration paths in the three saline habitats were different and all had specific water movement patterns. In the mild habitat, the water movement path in plants followed as: shallow root → stem → branches → leaves and shallow root → deep root. In moderate habitats, stems acted as the bifurcation point where the water movement path followed as: stem→ branches → leaves and stem → shallow root → deep root. In

severe habitats, the water movement path went from stem to shallow root. These water movement paths occur in the plants' xylem, not the epidermal cells, this may be because the photosynthetic organs of *Halostachys caspica* have the ability to absorb the condensation water in the canopy and transport the water to the xylem [18]. Another discovery was made in the study of the *Haloxylon ammodendron*. When the photosynthetic organs of the *Haloxylon ammodendron* absorb the water from the canopy, whose water potential keeps increasing, and when the water potential of photosynthetic organ increased to a certain degree, which may establish the reverse water potential gradient, namely Ψ Photosynthetic organs > Ψ Secondary branches, the photosynthetic organs can transfer excess water to the main stem via a reverse water potential gradient, and this is conducive to the continuous absorption and utilization of the condensation water in the canopy, it can be seen that the reverse water potential gradient is the energy structure of plant photosynthetic organs capable of absorbing the condensation water in the canopy [18].

It is clear that the water movement path in the mild habitat begins in the shallow root, while the path in the moderate and severe habitats begin in the stems. Thus, water migration paths start in different parts of the plant depending on the degree of salinity in the habitat. This may be related to the salt and water potential differences in plant structures. Previous studies have shown that hydrogen isotopic fractionation in drought-resistant, salt-tolerant, woody plants is likely to occur when water is being absorbed in the roots [45]. Thus, it seems this will also affect condensation water use by *Halostachys caspica*.

To summarize, condensation water absorption and utilization in the study plants took two forms: (1) Direct absorption where leaves directly absorbed condensation water, which can add moisture to the plant surface, reduce the surface temperature of leaves, and reduce the water loss of surface evaporation [46], or replenish the evaporation consumption, as well as go to the plant body for growth; and (2) Indirect absorption, where atmospheric condensation in the soil constitutes the soil water, while the atmospheric condensation water replenishes the river and groundwater that plant roots absorb in addition to soil water. These results reflect the findings of Goebel and Lascano [17], who measured the δD and δ $^{18}$O composition of different water sources of cotton and quantitatively analyzed the water content, including condensation water. Chen et al. [47] also conducted a study on 50 plant species in arid and semi-arid regions in Central Ningxia Province and found that the plant leaves could absorb water and even had the ability to take advantage of small amounts of rainfall.

## 5. Conclusions

In conclusion, this study revealed that: (1) Scale-like leaves can actively absorb condensation water, the condensation water absorbed by leaves can supplement the water consumed by evaporation to a certain extent, and the contribution of condensation water to plant growth in severe salinity habitats is the greatest (11.13%); (2) The migration path of water movement in the three habitats followed two main paths: (a) rainwater and condensation water were recharged through soil to compensate for groundwater, while some groundwater compensated for river water, and these were in part returned to the atmosphere by soil evaporation and plant transpiration; and (b) rainwater and condensation water directly compensated for the river, such that plant roots not only absorbed river water but also groundwater and soil water to assist with plant growth; (3) In mild salinity habitats, water movement paths in plants followed as: shallow root → stem → branches → leaves; and shallow root → deep root; in moderate habitats, stems acted as the bifurcation point and the path of water followed as: stem → branches → leaves, as well as: stem → shallow root → deep root; in severe habitats, the water movement path followed as: deep root → shallow root → stem → branches → leaves and finally returning to the atmosphere; there is also a water circuit from stem to shallow root. Although there are clear condensation water movement paths, this remains to be studied further, specifically in the xylem of the plant.

**Author Contributions:** Conceptualization, L.Q., X.H. and G.L.; investigation: L.Q., X.H. and J.Y.; methodology, L.Q. and X.H.; software, L.Q. and X.H.; writing—original draft, L.Q., X.H. and J.Y.; writing—review and editing, L.Q. and X.H.; supervision, G.L.; funding acquisition, G.L., L.Q. and X.H. All authors have read and agreed to the published version of the manuscript.

**Funding:** This work was funded by the National Natural Science Foundation of China (Nos. 41571034, 32101360, 31760168, 31660120) and the Doctor Starts Project of Xinjiang University (202115120003).

**Institutional Review Board Statement:** Not applicable.

**Informed Consent Statement:** Not applicable.

**Data Availability Statement:** Not applicable.

**Acknowledgments:** We thank Jing Cao, Xiao Ying, Ting-Quan Wang in Key Laboratory of Oasis Ecology of Xinjiang University for their indispensable help in fieldwork.

**Conflicts of Interest:** The authors declare no conflict of interest.

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
