# Peer review of "Regulatory Control and the Effects of Condensation Water on Water Migration and Reverse Migration of Halostachys caspica (M.Bieb.) C.A.Mey. in Different Saline Habitats"

_forests, doi:10.3390/f13091442_

Round 1

Reviewer 1 Report

Although the manuscript is prepared on the important aspect of water movement in soil and plant. Still the introduction is lacking the clarity about the utilization of this aspect by other arid plant species. Is this study applicable to conserve water? It is well known fact that plants use majorly ground water, so no new aspect is presented here.

Also the authors have not measured any xylem or phloem florescence studies to exactly demarcate the path of water movement, it is defined based on just the isotopic content. Water potential of the leaves and stem could also have been measured to define the pressure under different saline conditions for degree of evapotranspiration.  The rate of transpiration would have been the clear way for water route and it should be correlated with water absorption.

The authors have studied only passive movement of water, but under saline conditions, active movement also takes place and that needs to be studied based on plant kinetics.

No plant growth parameters have been studied, this manuscript is just a measurement of hydrogen and oxygen isotopes which does not find any scientific application in changing climatic conditions. Plant growth is not only due to water movement but its absorption and conservation strategies define the growth and productivity.

The results should be revised.

Reviewer 2 Report

The manuscript is interesting  with a novel theme, it studies the migration path of water movement in these three communities with different salinities. However, it would be necessary to indicate the average salinity levels of the areas included in the study.

In figure 1 the quality of the image should be improved.

In the material and methods section, the subsection III. Water extraction and isotope determination should be expanded and better detail the method used.

In summary the design of the experiments and the results are correct but the way of presenting them needs to be improved.

Round 2

Reviewer 1 Report

The authors have included the suggestions.